# Recognition of a disulfiram ethanol reaction in the emergency department is not always straightforward

Kristof Segher[1], Liesbeth Huys[2], Tania Desmet[3], Evi Steen[4], Stefanie Chys[5], Walter Buylaert[3], Peter De Paepe[3]*

1 Department of Emergency Medicine, AZ Alma, Eeklo, Belgium, 2 Department of Pharmacy, Ghent University Hospital, Ghent, Belgium, 3 Department of Emergency Medicine, Ghent University Hospital, Ghent, Belgium, 4 Department of Emergency Medicine, AZ Sint-Jan, Brugge, Belgium, 5 Department of Emergency Medicine, Algemeen Stedelijk Ziekenhuis (ASZ), Aalst, Belgium

* tania.desmet@uzgent.be

**Data Availability Statement:** All relevant data are within the paper and its Supporting Information files.

**Funding:** The authors received no specific funding for this work.

## Abstract

### Objectives

Disulfiram is an adjunct in the treatment of alcohol use disorders, but case reports indicate that disulfiram ethanol reactions are not always recognized in the emergency department. Our first aim is to remind of this risk with two case reports of life-threatening reactions not immediately considered by the emergency physician. The second aim is to estimate the probability that a disulfiram reaction goes unrecognized with the use of a retrospective study of patients admitted to the emergency department.

### Methods

Clinical files of patients admitted between October 1, 2010 and September 30, 2014 to the emergency department were retrospectively screened for the key words "ethanol use" and "disulfiram". Their diagnoses were then scored by a panel regarding the probability of an interaction.

### Results

Seventy-nine patients were included, and a disulfiram-ethanol reaction was scored as either 'highly likely', 'likely' or 'possible' in 54.4% and as 'doubtful' or 'certainly not present' in 45.6% of the patients. The interrater agreement was 0.71 (95% CI: 0.64–0.79). The diagnosis was not considered or only after a delay in 44.2% of the patients with a 'possible' to 'highly likely' disulfiram interaction. One patient with a disulfiram overdose died and was considered as a 'possible' interaction.

### Discussion and conclusions

A disulfiram ethanol interaction can be life threatening and failure to consider the diagnosis in the emergency department seems frequent. Prospective studies with documentation of the intake of disulfiram and evaluation of the value of acetaldehyde as a biomarker are

**Competing interests:** The authors have declared that no competing interests exist.

**Abbreviations:** ED, Emergency Department; DIS, Disulfiram; DER, Disulfiram Ethanol Reaction.

needed to determine the precise incidence. Improving knowledge of disulfiram interactions and adequate history taking of disulfiram intake may improve the care for patients.

## Introduction

Disulfiram (DIS) is primarily used as an adjunct in the treatment of alcohol use disorders [1–4]. DIS and its active metabolite S-methyl N,N-diethylthiocarbamate sulfoxide irreversibly inhibit the enzyme aldehyde dehydrogenase which leads to the accumulation of acetaldehyde with effects known as a DIS ethanol reaction (DER) [5, 6]. Recovery of enzymatic activity depends on de novo aldehyde dehydrogenase synthesis that takes place in 6 or more days [7]. A DER may occur after even small quantities of alcohol which usually leads to an unpleasant reaction [3]. A DER has been described occasionally after ingestion of either food cooked in alcohol or alcohol-based sauces and following excessive use of alcohol-containing cosmetics [8, 9]. Inhalation of alcohol vapour from hand sanitizers may transiently produce ethanol levels that are high enough to cause a mild DER [9, 10]. The necessity for supervised ingestion and the goal of sustained abstinence instead of reduced drinking are advocated because the use of even small amounts of ethanol during therapy can present as an emergency [4]. A DER may indeed not just be unpleasant with symptoms such as flushing of the face, throbbing in the head and pulsating headache but can also induce alarming systemic effects [11, 12]. These may even be life threatening with e.g. hemodynamic shock [11, 13–16], hypotension [12, 11–26], ST-segment depression [11, 14, 20, 26–28], stroke [21], cardiovascular collapse [12, 16, 19], cardiogenic shock [29], cardiac arrhythmias [12, 19], myocardial ischemia [17, 23], myocardial injury [28], myocardial infarction [12, 19, 30–32], unconsciousness [12, 19], convulsions [12, 19], dyspnea [19], respiratory difficulties [12, 19] and bronchospasm [17, 24, 33]. It should be noted that some case reports of DER concern DIS overdoses [25, 29, 38] and that sudden death has been reported as a DER with high therapeutic doses [11, 17, 34–36] and in DIS intentional overdoses combined with ethanol [37, 38]. Patients with a DER may seek help in the ED as illustrated in many case reports and a DER may initially go unrecognized [14–16, 24, 29]. The aims of the present study are (1) to highlight the clinical picture of a severe DER admitted to an ED with two illustrative cases and (2) to estimate the probability that a DER goes unrecognized with the use of a retrospective study of patients admitted to the emergency department.

## Materials and methods

The two case reports were observed by the authors SC and ES respectively. The assessment of the diagnostic process of a DER was carried out in the ED of the University Hospital of Ghent (Belgium) with a census of about 33 000 patients annually. The electronic files of all patients admitted between October 1, 2010 and September 30, 2014 were retrospectively screened for the term DIS or the registered name Antabuse® based on the following data fields: reason for admission, home medication history, diagnosis and treatment. It was also noted whether ethanol intake was mentioned in the history upon admission. Furthermore, data on symptoms, vital parameters, ethanol and lactate concentrations were collected and outcome was also registered. For systolic and diastolic blood pressure the lowest values observed in the ED were used.

Based on these data the likelihood of a DER was assessed independently by a panel of 6 authors (KS, ES, SC, WB, TD and LH) using a scoring between 1 and 5, with 1 being 'most

likely', 2, 3, 4 and 5 being 'likely', 'possible', 'doubtful' and 'certainly not' respectively. This was done for each case individually and the medium scores were calculated. In advance, the panel members received a number of papers about symptoms and signs of a DER [11, 20–23, 25, 27–29, 33, 36–40]. Moreover, DER symptoms reported in at least two scientific papers or in at least one of two reference handbooks [11, 12] were summarized in a table.

Patients with a medium DER score between 1 and 3 and those with a score between 3 and 5 were considered as group A and B respectively.

Data were analyzed as frequencies (percentages) for categorical variables and as means (± standard deviations) for continuous variables. Independent t-test was used to compare continuous data between groups A and B. Fisher's Exact Test or Pearson Chi-Square Test was used for comparison of distribution between and within the groups for the diagnosis of DER, symptoms, vital parameters including cardiovascular collapse and outcome.

The inter-rater agreement of the probability of the diagnosis of a DER was assessed with a weighted kappa with quadratic weights.

For all analyses, a two-tailed significance level of $p < 0.05$ was used. The statistical analyses were performed with IBM SPSS 25.0 for Windows.

The study was approved by the ethical committees of the University Hospital of Ghent, the Jan Yperman hospital (Ieper) and the Algemeen Stedelijk Ziekenhuis (ASZ) hospital Aalst. All data were fully anonymized before being assessed. Consent for publication of raw data was not obtained. The requirement for informed consent was waived by the ethics committee.

## Results

### Case reports

**Case 1.**  A 43-year-old man with schizophrenia under treatment with haloperidol, risperidone, clorazepate and DIS at a dose of 400 mg/day was found comatose at home (Glasgow Coma Score: 3/15). He was pale with cold extremities, hypotensive (blood pressure: 50/20 mmHg) and hypothermic (32°C). His respiration was depressed with a pulse oximetry saturation of 60%. The patient was endotracheally intubated and ventilated with improvement of the oxygen saturation, but the hypotension persisted. On admission to the ED the arterial blood gas showed a metabolic acidosis (pH 7.19) with increased lactate 7.17 mmol/L (normal upper limit: 1.60 mmol/L) and an ethanolemia of 1.46 g/L. Serum creatinine was increased to 0.19 mmol/L (normal upper limit: 0.08 mmol/L). An ethylene glycol or methanol poisoning was initially suspected but serum concentrations later appeared to be negative and toxicological screening was positive for benzodiazepines only.

A chest X-ray and cerebral CT-scan were normal. A DER was diagnosed and the patient was treated with crystalloids, noradrenaline infusion and external warming resulting in improvement of the blood pressure. He developed a ventilator associated pneumonia treated with amoxicillin/clavulanic acid and rhabdomyolysis (CK 44 000 IU/L: normal upper limit: 190 IU/L) treated with fluids, alkalinisation and mannitol 15%. He made a good recovery, but his renal function was decreased with a creatine of 0.32 mmol/L (normal upper limit: 0.08 mmol/L) at the 10th day after admission.

**Case 2.**  A 49-year-old man with a history of severe alcohol abuse became unwell and was transported by emergency medical technicians to the ED at 3 pm. The family informed them that the patient was under treatment with DIS and had used 7 units of beer in the afternoon resulting in abdominal pain and thirst. Neither medication nor empty blisters were found at his home. Upon admission he showed a generalized erythema, decreased consciousness with a Glasgow Coma Score of 13/15 (Eye movement: 3, Verbal response: 4, Motor reaction: 6), tachycardia (125 beats/min) and a low blood pressure (70/30 mmHg). The first attending

physician, a 1[st] year trainee in emergency medicine, tentatively diagnosed an anaphylactic shock but asked advice from the consultant in emergency medicine. The latter confirmed the presence of an erythema and found a diffusely painful abdomen on palpation. Auscultation of heart and lungs was normal. The patient was tachypneic with a peripheral oxygen saturation of 95% at room air. An arterial blood gas showed a pH of 7.50 (normal values 7.35–7.45), a bicarbonate of 20.8 mmol/L (normal values: 22–26 mmol/L), a base excess of -1.3 (normal values –3.5 to +3.5), a $pCO_2$ of 27.1 mmHg (normal values: 35–45 mmHg), a $pO_2$ of 83.8 mmHg (normal values: 83–108 mmHg), and a lactate of 4.1 mmol/L (normal values: 0.9–1.7 mmol/L). Further laboratory data showed a blood glucose of 179 mg/dl (normal values: 74–106 mg/dl), a sodium concentration of 133 mmol/L (normal values:136–145 mmol/L) and potassium concentration of 3.8 mmol/L (normal values: 3.6–4.8 mmol/L). The ethanolemia was 0.89 g/L and the consultant in emergency medicine diagnosed a DER. Because severe hypotension persisted even after a fast fluid bolus infusion of one liter of a balanced crystalloid solution, intravenous noradrenaline was started under invasive blood pressure monitoring. Administration of fomepizole was considered but not deemed necessary as the blood pressure rapidly recovered in the intensive care unit where noradrenaline could be discontinued, and the erythema disappeared. The patient could be discharged from the intensive care unit after 24 hrs.

## Retrospective analysis of ED patients with a history of DIS treatment and ethanol use

During the study period, 79 patient records containing both the term Antabuse[®] or DIS together with a history of ethanol intake, were included. The demographics and scores for a DER as assessed by the panel are shown in Table 1. The mean age was 46.72 ± 10 years and male patients were overrepresented (59.5%).

A DER was considered by the panel as either 'highly likely', 'likely' or 'possible' in 43 patients (54.4%) (group A) and as 'doubtful' or 'certainly not' in 36 patients (45.6%) (group B) (Table 1). The estimated kappa of inter-rater agreement for the 5 classes was 0.71 (95% CI: 0.64–0.79).

The admitting clinician did not consider the diagnosis of a DER in 19 out of the 43 (44.2%) patients in group A (Table 2). In the subgroup of patients with a highly likely DER this figure was 29.4% and in one patient the diagnosis was made but only after a delay.

Table 3 summarizes the clinical characteristics, vital parameters, laboratory data, therapy and outcome in the two groups.

Flushing, decreased consciousness and nausea were significantly more frequently reported in group A than in group B (p<0.001, p = 0.016 and p = 0.017 respectively). Mean systolic and

**Table 1. Demographics and DER scores of patients with a history of DIS treatment and ethanol use admitted to the ED of the University Hospital of Ghent between October 1, 2010 and September 30, 2014.** Scores are given as the medium value and the range of the individual scores by the 6 panel members. Patients are grouped (A and B) according to the likelihood of a DER.

| | All patients | Group A | | | | Group B |
|---|---|---|---|---|---|---|
| Score DER | | Highly likely | Likely | Possible | Total | Doubtful or certainly not |
| Medium score | NA[a] | 1 | > 1 and ≤ 2 | > 2 and ≤ 3 | NA | > 3 and ≤ 5 |
| **Gender** | N[b] = 79 (%) | N = 17 (%) | N = 7 (%) | N = 19 (%) | N = 43 (%) | N = 36 (%) |
| Male | 47 (59.5) | 10 (58.8) | 5 (71.4) | 8 (42.1) | 23 (53.5) | 24 (66.7) |
| Female | 32 (40.5) | 7 (41.2) | 2 (28.6) | 11 (57.9) | 20 (46.5) | 12 (40.5) |
| **Age (years)** | 46.72 ± 10 | 47.7 ± 10.9 | 41.7 ± 13.8 | 46.5 ± 8.6 | 46.16 ± 10.43 | 47.4 ± 10 |

[a] NA: not applicable

[b] N: number of patients

**Table 2. Consideration of the diagnosis of DER by the treating physician in patients with a history of DIS treatment and ethanol use admitted to the ED of the University Hospital of Ghent between October 1, 2010 and September 30, 2014.**

| | Group A | | | p-value |
|---|---|---|---|---|
| | **Highly likely** | **Likely** | **Possible** | |
| **Diagnosis DER** | N[a] = 17 (%) | N = 7 (%) | N = 19 (%) | |
| Immediate | 11 (64.7) | 6 (85.7) | 6 (31.6) | 0.021[b] |
| Delayed | 1 (5.9) | 0 | 0 | |
| Not considered | 5 (29.4) | 1 (14.3) | 13 (68.4) | |

[a] N: number of patients

[b] Fisher's Exact Test

diastolic blood pressures were significantly lower in group A than in group B (p = 0.006 and p = 0.001 respectively). There were no differences between group A and B regarding heart rate and body temperature.

One case of cardiovascular collapse was observed (group A) in a 41 years old man who was found at home with decreased consciousness and a collateral history from relatives of an acute overdose with DIS and diazepam. Upon arrival of the ambulance, he still had respiratory activity, but the cardiac rhythm evolved to asystoly upon arrival of a Mobile Intensive Care Unit team.

During resuscitation, including external cardiac massage and endotracheal intubation which revealed aspiration of food, he developed a ventricular fibrillation. Following one direct current shock there was a return of spontaneous circulation. In the ED his laboratory data revealed a severe acidosis (pH of 7.0; normal values: 7.35–7.45), an increased lactate (7.28 mmol/L; normal upper limit: 1.60 mmol/L), renal insufficiency (creatinine 0.21 mmol/L; normal values: 0.064–0.103 mmol/L) and an ethanolemia of 0.9 g/L. A chest X-ray showed a pneumonia and he was treated in the ICU with artificial ventilation, antibiotics and hemodialysis. His cardiorespiratory and renal functions initially improved but he remained unresponsive and neurological examinations showed postanoctic damage. The patient eventually died 9 days after admission.

Regarding the therapy in our series of patients, fluid loading was significantly more frequent in group A (p = 0.002) with a vasopressor being needed in one patient of this group. Plasma lactate was available in only about 30% of all patients and was not significantly higher in group A than in B. Ethanolemia was available in 77% of the patients with a mean value that was higher in group B than in group A, but not significantly so.

Regarding the outcome most patients could either be discharged home or had to be admitted to a ward preceded by an ED observation in some cases. The discharge pattern was not different between group A and B.

## Discussion

DIS is mainly used in the pharmacotherapy of alcohol use disorders but also for the treatment of cocaine and other stimulant dependence [41–43]. More recent interest in disulfiram for treating various cancers has provided some renewed clinical interest [44–46]. A meta-analysis of the efficacy of disulfiram in treatment of alcohol dependence concluded that evidence from well-controlled trials does not adequately support an association with preventing return to any drinking or improvement in other alcohol consumption outcomes [2]. Moreover, experiencing a DER seems not to be associated with any differences in treatment outcome but with a significant earlier discontinuation of DIS therapy [47]. Other publications [1, 3, 4] concluded that

**Table 3. Clinical characteristics and outcome in patients with a history of DIS treatment and ethanol use admitted to the ED of the University Hospital of Ghent between October 1, 2010 and September 30, 2014.** Patients are grouped (A and B) according to the likelihood of a DER as attributed by the panel.

| | All patients | Group A | | | | Group B | P value |
|---|---|---|---|---|---|---|---|
| | | Highly likely | Likely | Possible | Total | Doubtful or certainly not | |
| **Symptoms** | N = 79 (%) | N = 17 (%) | N = 7 (%) | N = 19 (%) | N = 43 (%) | N = 36 (%) | |
| Confusion | 18 (22.8) | 4 (23.5) | 1 (14.3) | 7 (36.8) | 12 (27.9) | 5 (13.9) | > 0.05[a] |
| Flushing | 14 (17.7) | 11 (64.7) | 1 (14.3) | 2 (10.5) | 14 (32.6) | 0 | < 0.001[a] |
| Decreased consciousness | 14 (17.7) | 6 (35.3) | 3 (42.9) | 3 (15.8) | 12 (27.9) | 2 (5.6) | 0.016[a] |
| Nausea | 13 (16.5) | 7 (41.2) | 1 (14.3) | 3 (15.8) | 11 (25.6) | 2 (5.6) | 0.017[a] |
| Vomiting | 10 (12.7) | 3 (17.6) | 1 (14.3) | 3 (15.8) | 7 (16.3) | 3 (8.3) | > 0.05[b] |
| Palpitations | 6 (7.6) | 2 (11.8) | 0 | 3 (15.8) | 5 (11.6) | 1 (2.8) | > 0.05[b] |
| Dyspnoe | 6 (7.6) | 2 (11.8) | 0 | 3 (15.8) | 5 (11.6) | 1 (2.8) | > 0.05[b] |
| Retrosternal pain | 5 (6.3) | 3 (11.8) | 0 | 2 (10.5) | 4 (9.3) | 1 (2.8) | > 0.05[b] |
| Headache | 5 (6.3) | 1 (5.9) | 2 (28.6) | 1 (5.3) | 4 (9.3) | 1 (2.8) | > 0.05[b] |
| Tremor | 5 (6.3) | 1 (5.9) | 2 (28.6) | 1 (5.3) | 4 (9.3) | 1 (2.8) | > 0.05[b] |
| Abdominal pain | 5 (6.3) | 0 | 1 (14.3) | 1 (5.3) | 2 (4.7) | 3 (8.3) | > 0.05[b] |
| Epileptic insult | 4 (5.1) | 0 | 1 (14.3) | 1 (5.3) | 2 (4.7) | 2 (5.6) | > 0.05[b] |
| Vertigo | 3 (3.8) | 0 | 1 (14.3) | 2 (10.5) | 3 (7) | 0 | > 0.05[b] |
| Diaphoresis | 2 (2.5) | 1 (5.9) | 0 | 0 | 1 (2.3) | 1 (2.8) | > 0.05[b] |
| Pruritus | 1 (1.3) | 0 | 0 | 1 (5.3) | 1 (2.3) | 0 | > 0.05[b] |
| Myalgia | 1 (1.3) | 0 | 0 | 1 (5.3) | 1 (2.3) | 0 | > 0.05[b] |
| Hyperventilation | 1 (1.3) | 0 | 0 | 1 (5.3) | 1 (2.3) | 0 | > 0.05[b] |
| Respiratory difficulties/ depression | 1 (1.3) | 0 | 0 | 0 | 0 | 1 (2.8) | > 0.05[b] |
| **Vital parameters** | N = 78 (1 missing value) | N = 17 | N = 7 | N = 19 | N = 43 | N = 35 (1 missing value) | |
| Systolic blood pressure (mmHg) | 101.99 ± 21.18 | 92.06 ± 4.8 | 78 ± 8.36 | 106.47 ± 4.67 | 96.14 ± 22.66 | 109.17 ± 2.85 | 0.006[c] |
| Diastolic blood pressure (mmHg) | 59.59 ± 17.33 | 49.65 ± 3.53 | 43.57 ± 6.03 | 61.79 ± 3.48 | 54.02 ± 16.39 | 66.43 ± 2.73 | 0.001[c] |
| | N = 79 (%) | N = 17 (%) | N = 7 (%) | N = 19 (%) | N = 43 (%) | N = 36 (%) | |
| Cardiovascular collapse | 1 (1.3) | 0 | 0 | 1 (5.3) | 1 (2.3) | 0 | > 0.05[b] |
| | N = 77 (2 missing values) | N = 17 | N = 7 | N = 18 (1 missing value) | N = 42 (1 missing value) | N = 35 (1 missing value) | |
| Heart rate (beats/minute) | 105.75 ± 19.27 | 112.82 ± 4.12 | 111.71 ± 5.31 | 103.78 ± 5.31 | 108.76 ± 19.26 | 102.14 ± 3.2 | > 0.05[c] |
| | N = 66 (13 missing values) | N = 14 (3 missing values) | N = 7 | N = 16 (3 missing values) | N = 37 (6 missing values) | N = 29 (7 missing values) | |

(*Continued*)

**Table 3.** (Continued)

| | All patients | Group A | | | | Group B | P value |
|---|---|---|---|---|---|---|---|
| | | Highly likely | Likely | Possible | Total | Doubtful or certainly not | |
| Temperature (˚C) | 36.35 ± 0.94 | 36.24 ± 0.2 | 36.68 ± 0.19 | 36.12 ± 0.39 | 36.27 ± 1.14 | 36.45 ± 0.11 | > 0.05[c] |
| **Ethanolemia (g/l)** | N = 70 (9 missing values) | N = 15 (2 missing values) | N = 7 | N = 16 (3 missing values) | N = 38 (5 missing values) | N = 32 (4 missing values) | |
| | 1.82 ± 1.17 | 1.31 ± 0.21 | 1.14 ± 0.55 | 2.08 ± 0.22 | 1.60 ± 1.04 | 2.07 ± 0.23 | > 0.05[c] |
| **Lactate (mmol/l)** | N = 24 (55 missing values) | N = 7 (10 missing values) | N = 2 (5 missing values) | N = 6 (13 missing value) | N = 15 (28 missing value) | N = 9 (27 missing values) | |
| | 4.03 ± 3.20 | 4.44 ± 2.34 | 3.50 ± 2.99 | 4.87 ± 1.96 | 4.48 ± 2.14 | 3.27 ± 4.51 | > 0.05[c] |
| **Therapy** | N = 79 (%) | N = 17 (%) | N = 7 (%) | N = 19 (%) | N = 43 (%) | N = 36 (%) | |
| Fluid loading | 37 (46.8) | 13 (76.5) | 6 (85.7) | 8 (42.1) | 27 (62.8) | 10 (27.8) | 0.002[a] |
| Vasopressors | 1 (1.3) | 0 | 0 | 1 (5.3) | 1 (2.3) | 0 | > 0.05[b] |
| **Outcome** | N = 78 (%) (1 missing value) | N = 17 (%) | N = 7 (%) | N = 18 (%) (1 missing value) | N = 42 (%) (1 missing value) | N = 36 (%) | |
| ED observ. + disch. to psychiatry | 28 (35.9) | 6 (35.3) | 2 (28.6) | 5 (27.8) | 13 (31) | 15 (41.7) | > 0.05[b] |
| ED observation and discharge home | 27 (34.6) | 7 (41.2) | 5 (71.4) | 7 (38.9) | 19 (34.6) | 8 (22.2) | |
| Discharge to psychiatry | 9 (11.5) | 3 (17.6) | 0 | 1 (5.6) | 4 (9.5) | 5 (13.9) | |
| ED observ. and discharge to a ward | 6 (7.7) | 0 | 0 | 2 (11.1) | 2 (4.8) | 4 (11.1) | |
| Discharge home | 5 (6.4) | 1 (5.9) | 0 | 2 (11.1) | 3 (7.1) | 2 (5.6) | |
| ICU | 2 (2.6) | 0 | 0 | 0 | 0 | 2 (5.6) | |
| Death | 1 (1.3) | 0 | 0 | 1 (5.6) | 1 (2.4) | 0 | |

N: number of patients

[a] Pearson Chi Square test

[b] Fisher's Exact Test

[c] Independent T-test

the drug is valuable but in these, adequate supervision was guaranteed. In this respect immediate recognition and treatment of a DER in the ED is an important aspect of DIS therapy. These reactions are indeed sometimes life threatening and require adequate therapy. Moreover, these DER's in the ED can offer valuable feedback to the treating physician as they signal problems with abstinence from ethanol.

The first aim of our study was to remind clinicians of these severe reactions with two illustrative case reports observed in the ED. The first patient developed renal insufficiency (presumably resulting from a combination of hypoperfusion and rhabdomyolysis) and both cases showed severe hypotension and are highly suggestive for a DER. Importantly both cases also illustrate that the recognition of a DER is not always immediately made by the clinicians which can lead to a delayed or even missed diagnosis [14–16, 29].

As far as we know there are no systematic studies on the frequency and the detection of ED admissions for DER. Therefore, the second aim of this paper was to study this aspect by analyzing the clinical files for the presence of a DER in all patients admitted to an ED with a history of DIS treatment and ethanol intake. In this retrospective study, 79 patient records contained the term DIS and mentioned ethanol use representing about 20 patients per year.

It should be noted however, that this figure may be an underestimate since medication history by the emergency physician may have been incomplete as shown by studies in which a pharmacist actively was involved in the medication history [48].

Moreover, due to the irreversible inhibition of the enzyme aldehyde dehydrogenase, which leads to a prolonged accumulation of acetaldehyde, the history should also take a recent stop of DIS into account. A false feeling of security may be present in patients who recently stopped the intake, and this may lead them not to mention this to the admitting clinician. A prospective study with a thorough medication history would shed more light on the incidence of potential DER's in the ED.

Because the diagnosis of a DER in a retrospective study of routine clinical practice is relying on the interpretation of clinical data, we analyzed the clinical files with a panel composed of emergency physicians and a clinical pharmacist. This analysis was done independently by each member to avoid mutual influence and was preceded by providing literature data on DER's including a case of DIS overdose. This may have increased the awareness of a DER in the panel members and the likelihood of ascribing a picture of clinical aspects to a DER. Patients with signs such as flushing, decreased consciousness, nausea, hypotension and a need for fluid therapy were significantly more frequently categorized by the panel in group A than group B. This is presumably explained by the fact that these are indeed well-known features of a DER.

The kappa value for the inter-rater variability of 0.71 can be considered as moderate [49] to substantial [50]. This rather large interobserver variability may be explained by the fact that a DER remains a clinical diagnosis.

In 45.6% of the patients a DER was scored as 'doubtful' or 'certainly not present' (group B). Five patients in this group had a negative ethanolemia which contrasts with only one patient in group A. This negative ethanolemia may have contributed to the assignment by the panel of patients to group B.

Besides the negative ethanolemia other explanations for the assignment of patients to group B should also be considered. First the clinical signs of a DER have been described to be much weaker in patients with alcoholic liver disease than in those without [51]. Therefore, in future studies of DER's in the ED it would be of interest to study the presence of alcoholic liver disease.

A second explanation may be that patients in group B were less adherent to their DIS therapy and had not taken DIS in the two weeks prior to admission. A third explanation may be that the clinical information in the files of group B patients was more frequently incomplete or inaccurate because the admitting clinician implicitly ascribed DER symptoms to ethanol poisoning alone. The fact that the mean ethanolemia in group B is higher than in group A favors this hypothesis. Future prospective studies with a focus on accurate and complete DIS medication history and determining blood concentrations, use of a checklist of signs of DER by clinicians and assessment of the presence of alcoholic liver disease would give better insight into the real incidence of DER's in the ED. Finally, a fourth explanation for the assignment of patients to group B may be that clinical symptoms and acetaldehyde plasma concentrations under treatment with DIS appear to decrease after repetitive exposure to ethanol [52].

In view of the uncertainties in the clinical evaluation, a biochemical indicator would be of interest to objectivate which patients suffered from a DER. In our study plasma lactate seemed higher in group A than in group B (not statistically significant) and may also have been considered by the panel members as an argument for diagnosing a DER. A high number of lactate values were missing, which could explain why the lactate level was not significantly higher in group A. A systematic study of plasma lactate in patients with a possible DER is needed to evaluate its value as a biochemical marker and whether it adds further information to the parameter of hypotension.

In addition to plasma lactate, measurement of acetaldehyde in plasma or red blood cells may be of interest as reported in some case reports [14]. Plasma concentrations of acetaldehyde increase during ethanol challenge in volunteers treated with DIS [53]. However, further studies are required to assess the sensitivity and specificity of acetaldehyde in the diagnosis of DER as plasma and red blood cell acetaldehyde already increase in abstinent alcoholic patients treated with DIS [54] and the increase in acetaldehyde appears to diminish after repeated exposure to ethanol [52]. The relationship of acetaldehyde levels with a DER is not fully understood [55]. Determination of the presence of DIS and its active metabolites in plasma may be considered to document a DER but one should be aware that the enzyme inhibiting effect of these compounds may still be observed following their disappearance as they are considered irreversible inhibitors [56–58].

Besides documenting a DER, monitoring retrospective alcohol use in patients on DIS treatment may be useful to verify abstinence. In this context, urinary ethylglucuronide (a breakdown product of ethanol) was shown to be a promising biomarker of ethanol exposure as it can be detected up to 5 days after drinking alcohol [59].

An important finding in our study is that the admitting emergency physicians did not consider the diagnosis of a DER even in patients for which the panel scored a DER as highly likely. Also, in the highly likely category, the diagnosis of DER in one patient was made only after a delay. Likewise, such a delay also occurred in the presented case reports. In the first case report the clinician initially suspected methanol or ethylene glycol poisoning and in the second case there was a temporary misinterpretation of DER symptoms as an anaphylactic reaction. These diagnostic failures or delays, which are also regularly reported in the literature [14–16, 29, 37] may have important consequences. Unnecessary diagnostic investigations can be avoided as well as the futile use of dopamine as a vasopressor as it is ineffective due to the inhibition of dopamine betahydroxylase by DIS [13, 22]. Moreover, in some resistant cases of DER, specific therapy with fomepizole should be considered.

Fomepizole, an inhibitor of alcohol dehydrogenase, limits the progression of the DER by blocking ethanol metabolism to acetaldehyde [17, 26, 27, 60]. Therefore, clinicians should be better aware of DER's and consider these reactions especially in the differential diagnosis of patients with features of e.g. an anaphylactic shock or distributive shock. Finally, it obviously is important that DER's are recognized and reported to the physician treating the alcohol use disorder to direct further therapy.

Regarding the outcome, most patients needed at least observation in the ED followed by admission to a ward which implicates a considerable workload and cost. One patient in our retrospective series ultimately died and it is important to note that according to his sister he took an overdose of DIS.

High doses, as used in older treatment regimens, have been described to lead to more severe DER's [34] and several cases of a life threatening [24–25, 38] and even fatal DER [37] have been described after DIS overdose. Although we did not dose DIS, this fatal case in our study was considered by the panel as a possible DER. The panel probably assumed that the patient died because of the decreased consciousness with vomiting that resulted in hypoxia with cardiac arrest leading to irreversible brain damage. Clinicians admitting patients with a DIS overdose should also be aware of this increased risk of a severe DER.

There are some limitations to this study. We may have underestimated the incidence of DER's in the ED because of the retrospective nature of this research. This may be due to incomplete medication history of DIS use and the failure to recognize and register symptoms and disturbances related to a DER. Furthermore, the diagnosis was made on clinical grounds as in routine clinical practice and the differential diagnosis was not always elaborated to exclude other causes for the symptoms.

Future prospective studies with adequate history taking are therefore necessary to determine the real incidence of a DER. Determining blood concentrations of DIS or its more easily detected metabolites like carbon disulphide, could document the intake of DIS and the utility of acetaldehyde measurements should be assessed.

## Conclusion

In summary, the present study illustrates that a DER can be severe and even life threatening and that the diagnosis is not readily considered or may be delayed in the ED. This can have important consequences regarding the urgent treatment of these patients and for the feedback to the clinicians supervising the treatment with DIS. Enhancing awareness of the signs and symptoms of a DER by education combined with more attention for an adequate history of ethanol use and DIS therapy, also when recently stopped, may improve the care of these patients. Future prospective studies to determine the precise incidence of DER in the ED and to explore the diagnostic value of biochemical markers are necessary.

## Supporting information

**S1 File.**
(XLSX)

## Acknowledgments

The authors thank Roos Colman of the department of statistics, Ghent University, for the statistical advice.

Part of the data were presented at the European Association of Poisons Centres and Clinical Toxicologists (EAPCCT) meeting in Malta (May 29, 2015) and one case was presented at the meeting of the Belgian Society for Emergency and Disaster Medicine (Besedim) (January 17, 2015).

## Author Contributions

**Conceptualization:** Peter De Paepe.

**Data curation:** Kristof Segher, Liesbeth Huys, Tania Desmet, Evi Steen, Stefanie Chys.

**Formal analysis:** Kristof Segher, Liesbeth Huys, Walter Buylaert.

**Funding acquisition:** Walter Buylaert, Peter De Paepe.

**Investigation:** Kristof Segher, Liesbeth Huys, Peter De Paepe.

**Methodology:** Kristof Segher, Liesbeth Huys, Peter De Paepe.

**Project administration:** Peter De Paepe.

**Resources:** Walter Buylaert, Peter De Paepe.

**Software:** Kristof Segher, Liesbeth Huys, Walter Buylaert.

**Supervision:** Peter De Paepe.

**Validation:** Kristof Segher, Liesbeth Huys, Peter De Paepe.

**Visualization:** Kristof Segher, Liesbeth Huys, Peter De Paepe.

**Writing – original draft:** Kristof Segher, Liesbeth Huys.

**Writing – review & editing:** Liesbeth Huys, Tania Desmet, Peter De Paepe.

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
