## [Decision Letter · Decision Letter 0]

7 Sep 2020

PONE-D-20-14245

Recognition of a disulfiram ethanol reaction in the emergency department is not always straightforward.

PLOS ONE

Dear Dr. Desmet,

Thank you for submitting your manuscript to PLOS ONE; I apologise for the unusually delayed review timeframe. After careful consideration, we feel that it has merit but does not fully meet PLOS ONE’s publication criteria as it currently stands. Therefore, we invite you to submit a revised version of the manuscript that addresses the points raised during the review process.

Your manuscript has been assessed by two reviewers, whose comments are appended below. In addition to addressing the concerns that they have raised, please ensure that you clarify whether the records used in this study were anonymised and/or whether the IRB waived the requirement for informed consent. Furthermore, we note that one or more reviewers has recommended that you cite specific previously published works. As always, we recommend that you please review and evaluate the requested works to determine whether they are relevant and should be cited. It is not a requirement to cite these works.

We look forward to receiving your revised manuscript.

Kind regards,

Emily Chenette

Deputy Editor-in-Chief

PLOS ONE

Journal Requirements:

2. In the ethics statement in the manuscript and in the online submission form, please provide additional information about the patient records used in your retrospective study. Specifically, please ensure that you have discussed whether all data were fully anonymized before you accessed them and/or whether the IRB or ethics committee waived the requirement for informed consent. If patients provided informed written consent to have data from their medical records used in research, please include this information.

3.We note that you have indicated that data from this study are available upon request. PLOS only allows data to be available upon request if there are legal or ethical restrictions on sharing data publicly. For information on unacceptable data access restrictions, please see http://journals.plos.org/plosone/s/data-availability#loc-unacceptable-data-access-restrictions.

Reviewers' comments:

Reviewer's Responses to Questions

**Comments to the Author**

1. Is the manuscript technically sound, and do the data support the conclusions?

Reviewer #1: Yes

Reviewer #2: Yes

2. Has the statistical analysis been performed appropriately and rigorously? 

Reviewer #1: Yes

Reviewer #2: I Don't Know

3. Have the authors made all data underlying the findings in their manuscript fully available?

Reviewer #1: Yes

Reviewer #2: Yes

4. Is the manuscript presented in an intelligible fashion and written in standard English?

Reviewer #1: Yes

Reviewer #2: Yes

5. Review Comments to the Author

Reviewer #1: This is a clinically important piece of work that points to a relevant problem and investigates it in an interesting, intelligent way. Only some important literary quotations are still missing from my point of view:

1) Page 13, line 222: Disulfiram is also increasingly used for the following conditions: cocaine addiction, tumor diseases and pathological gambling. Please cite papers for each disease. This makes the examination all the more relevant.

2) Page 15, line 283: There are biochemical markers, which can improve and verify a hidden DSR, please cite this paper and discuss it: Urinary Ethylglucuronide Assessment in Patients Treated With Disulfiram: A Tool to Improve Verification of Abstention and Safety, Mutschler et al, Clin Neuropharmacol . Nov-Dec 2010;33(6):285-7. doi: 10.1097/WNF.0b013e3181fc9362.

3) Diskussion: I think it would be also interesting to short discuss this paper by Mutschler et al: Experienced Acetaldehyde Reaction Does Not Improve Treatment Response in Outpatients Treated With Supervised Disulfiram, Clin Neuropharmacol . Jul-Aug 2011;34(4):161-5. doi: 10.1097/WNF.0b013e3182216fd5.

4) It would be interesting to discuss the effect of the drug in more detail (biological effect, psychological effect). Furthermore pharmacogenetic aspects.

Reviewer #2: This study provides a new perspective on the difulfiram-ethanol reaction (DER) and is useful to emergency room clinicians. Assuming the statistical analysis to be adequate and appropriate, the main body of the research seems to need no modification. However, a few peripheral points would benefit from clarification or expansion.

1. Describing the DER as an ‘aversive reaction’ tout court reinforces the widespread but incorrect notion that disulfiram (DSF) works by ‘aversion’ – i.e. the repeated coupling of drinking with an unpleasant response. In reality, most patients never deliberately test out DSF’s acataldehydaemic potential because the vicarious knowledge of that potential deters them from drinking while ALDH inhibition persists. The mechanism is deterrence, not aversion.

2. DSF and its active metabolite S-methyl N,Ndiethylthiolcarbamate

sulfoxide do not have long half-lives and are usually undetectable after 48 hours. However, because it is an irreversible ‘suicide inhibitor’ of ALDH, ALDH inhibition persists until the body produces new ALDH in adequate quantities. This process is probably genetically determined and can be as short as a couple of days or as long as 10 days – occasionally more.

3. Carbon disulphide – another metabolite of DSF – may be easier to detect than DSF itself. It, too, persists for about two days after the last dose.

4. DSF dosage information was not systematically collected in the study but one patient is recorded as having 400mg daily. Although bioavailability can vary with the manufacturing process, a starting dose of 200-250-mg is usually appropriate, though some supervised DSF programmes use 400/400/5-600mg on Monday/Wednesday/Friday. Dosage will only need increasing if the patient risks drinking and gets no reaction or only a very mild one.

5. Although I have never had occasion to use fomepizole, I think it deserves a slightly longer discussion, including its mode of action. I also think there may be a case for using it in any severe case of suspected DER pending further investigations or confirmation.

6. The authors say that even small quantities of alcohol can cause a DER. ‘Small’ is a rather elastic but in reality, small amounts such as in sauces never cause a reaction, or only an extremely mild one, though patients who inadvertently swallow small amounts of an alcoholic sauce sometimes have a panic attack that can superficially resemble a DER. Some very recent papers have drawn attention to the fact that inadvertently inhaling alcohol in hand-sanitisers can cause a DER, probably because of a rapid but transient build-up of acetaldehyde in lung tissues that then passes directly and quickly to the heart and brain. While COVID persists, this may be worth mentioning. However, the total amount of alcohol inhaled – compared with the amount in even one glass of wine – is so small that severe DERs from this route seem very unlikely and alcohol absorption directly through the skin is negligible.

6. PLOS authors have the option to publish the peer review history of their article (what does this mean?). If published, this will include your full peer review and any attached files.

Reviewer #1: No

Reviewer #2: **Yes: **Colin Brewer

---

## [Author Response · Author response to Decision Letter 0]

12 Oct 2020

Dear Editor-in-chief,

Dear Reviewers,

First of all, we would like to thank you and the reviewers for the time spent on reviewing the manuscript and the thoughtful comments helping us improving the article. In the attached letter we would like to answer the comments and questions listed.

Reviewer #1

1. Page 13, line 222: Disulfiram is also increasingly used for the following conditions: cocaine addiction, tumor diseases and pathological gambling. Please cite papers for each disease. This makes the examination all the more relevant.

 Other conditions in which disulfiram is used for (including references) have been added to the manuscript (line 229-231). 

2. Page 15, line 283: There are biochemical markers, which can improve and verify a hidden DSR, please cite this paper and discuss it: Urinary Ethylglucuronide Assessment in Patients Treated With Disulfiram: A Tool to Improve Verification of Abstention and Safety, Mutschler et al, Clin Neuropharmacol . Nov-Dec 2010;33(6):285-7. doi: 10.1097/WNF.0b013e3181fc9362.

 The authors thank you for this contribution and have adapted the text accordingly (line 313- 316).

 

3. Diskussion: I think it would be also interesting to short discuss this paper by Mutschler et al: Experienced Acetaldehyde Reaction Does Not Improve Treatment Response in Outpatients Treated With Supervised Disulfiram, Clin Neuropharmacol . Jul-Aug 2011;34(4):161-5. doi: 10.1097/WNF.0b013e3182216fd5.

 The authors thank you for this addition and have incorporated this interesting observation in the manuscript (line 234-235 ; reference 47).

4. It would be interesting to discuss the effect of the drug in more detail (biological effect, psychological effect). Furthermore pharmacogenetic aspects.

 The authors agree with these comments. The manuscript has been adapted based on the suggestions provided.

 Although this is a very interesting comment, we feel that a discussion on the biological and psychological effects is beyond the scope of this paper. We would therefore suggest not including it. It is not clear to us to which specific pharmacogenetic aspects of disulfiram the reviewer is alluding. 

Reviewer #2

1. Describing the DER as an ‘aversive reaction’ tout court reinforces the widespread but incorrect notion that disulfiram (DSF) works by ‘aversion’ – i.e. the repeated coupling of drinking with an unpleasant response. In reality, most patients never deliberately test out DSF’s acataldehydaemic potential because the vicarious knowledge of that potential deters them from drinking while ALDH inhibition persists. The mechanism is deterrence, not aversion.

 The authors agree with these comments. The manuscript has been adapted and the term « aversive reaction » has been omitted throughout the entire manuscript.

2. DSF and its active metabolite S-methyl N,Ndiethylthiolcarbamate sulfoxide do not have long half-lives and are usually undetectable after 48 hours. However, because it is an irreversible ‘suicide inhibitor’ of ALDH, ALDH inhibition persists until the body produces new ALDH in adequate quantities. This process is probably genetically determined and can be as short as a couple of days or as long as 10 days – occasionally more.

 The authors thank you for this contribution and have adapted the text accordingly (line 62-66 & line 308-311). 

3. Carbon disulphide – another metabolite of DSF – may be easier to detect than DSF itself. It, too, persists for about two days after the last dose.

 The authors thank you for this contribution and have adapted the text accordingly (line 352-353).

4. DSF dosage information was not systematically collected in the study but one patient is recorded as having 400mg daily. Although bioavailability can vary with the manufacturing process, a starting dose of 200-250-mg is usually appropriate, though some supervised DSF programmes use 400/400/5-600mg on Monday/Wednesday/Friday. Dosage will only need increasing if the patient risks drinking and gets no reaction or only a very mild one.

 The dosage of disulfiram was not systematically reported in this study. This fell beyond the scope of the study. The authors agree that a gradual increase of the dosage of disulfiram may be preferable. 

5. Although I have never had occasion to use fomepizole, I think it deserves a slightly longer discussion, including its mode of action. I also think there may be a case for using it in any severe case of suspected DER pending further investigations or confirmation.

 The authors agree with these comments. The manuscript has been adapted based on the suggestions provided (line 328-329).

6. The authors say that even small quantities of alcohol can cause a DER. ‘Small’ is a rather elastic but in reality, small amounts such as in sauces never cause a reaction, or only an extremely mild one, though patients who inadvertently swallow small amounts of an alcoholic sauce sometimes have a panic attack that can superficially resemble a DER. Some very recent papers have drawn attention to the fact that inadvertently inhaling alcohol in hand-sanitisers can cause a DER, probably because of a rapid but transient build-up of acetaldehyde in lung tissues that then passes directly and quickly to the heart and brain. While COVID persists, this may be worth mentioning. However, the total amount of alcohol inhaled – compared with the amount in even one glass of wine – is so small that severe DERs from this route seem very unlikely and alcohol absorption directly through the skin is negligible.

 The authors thank you for this addition and have incorporated this interesting observation in the manuscript (line 66-70).

Kindest regards,

Dr. Tania Desmet

---

## [Decision Letter · Decision Letter 1]

27 Oct 2020

PONE-D-20-14245R1

Recognition of a disulfiram ethanol reaction in the emergency department is not always straightforward.

PLOS ONE

Dear Dr. Desmet,

Thank you for submitting your manuscript to PLOS ONE. Thank you for submitting the revised manuscript. There were minor revisions recommended by one of our reviewers. Before we can accept this manuscript for publication, we ask you to consider addressing the reviewer's comments.

We look forward to receiving your revised manuscript.

Kind regards,

Steve Lin

Academic Editor

PLOS ONE

Reviewers' comments:

Reviewer's Responses to Questions

**Comments to the Author**

1. If the authors have adequately addressed your comments raised in a previous round of review and you feel that this manuscript is now acceptable for publication, you may indicate that here to bypass the “Comments to the Author” section, enter your conflict of interest statement in the “Confidential to Editor” section, and submit your "Accept" recommendation.

Reviewer #1: All comments have been addressed

Reviewer #2: (No Response)

2. Is the manuscript technically sound, and do the data support the conclusions?

Reviewer #1: Yes

Reviewer #2: Yes

3. Has the statistical analysis been performed appropriately and rigorously? 

Reviewer #1: Yes

Reviewer #2: I Don't Know

4. Have the authors made all data underlying the findings in their manuscript fully available?

Reviewer #1: Yes

Reviewer #2: Yes

5. Is the manuscript presented in an intelligible fashion and written in standard English?

Reviewer #1: Yes

Reviewer #2: Yes

6. Review Comments to the Author

Reviewer #1: I suggest the publication of this clinically important paper.

..............................................................................................................................................................

Reviewer #2: I am sorry if I seem difficult to satisfy but despite your agreement that the term 'aversive' is inappropriate, it still appears in Line 67, Please replace it with 'unpleasant'.

Similarly, you have again claimed a long half-life for disulfiram, despite accepting that it is the effect on ALDH that is prolonged, not the half-life. Please correct this.

7. PLOS authors have the option to publish the peer review history of their article (what does this mean?). If published, this will include your full peer review and any attached files.

Reviewer #1: No

Reviewer #2: **Yes: **Dr Colin Brewer

---

## [Author Response · Author response to Decision Letter 1]

2 Nov 2020

Dear Editor-in-chief,

Dear Reviewers,

First of all, we would like to thank you and the reviewers for the time spent on reviewing the manuscript and the thoughtful comments helping us improving the article. In the attached letter we would like to answer the comments and questions listed.

Reviewer #2

1. I am sorry if I seem difficult to satisfy but despite your agreement that the term 'aversive' is inappropriate, it still appears in Line 67, Please replace it with 'unpleasant'.

 The authors thank you for this contribution and have adapted the text accordingly (line 67).

2. Similarly, you have again claimed a long half-life for disulfiram, despite accepting that it is the effect on ALDH that is prolonged, not the half-life. Please correct this.

 The authors thank you for this contribution and have adapted the text accordingly (line 258- 259).

Kindest regards,

Dr. Tania Desmet

---

## [Editor Report · Decision Letter 2]

18 Nov 2020

Recognition of a disulfiram ethanol reaction in the emergency department is not always straightforward.

PONE-D-20-14245R2

Dear Dr. Desmet,

We’re pleased to inform you that your manuscript has been judged scientifically suitable for publication and will be formally accepted for publication once it meets all outstanding technical requirements.

Kind regards,

Steve Lin

Academic Editor

PLOS ONE
---

## [Editor Report · Acceptance letter]

23 Nov 2020

PONE-D-20-14245R2 

Recognition of a disulfiram ethanol reaction in the emergency department is not always straightforward 

Dear Dr. Desmet:

I'm pleased to inform you that your manuscript has been deemed suitable for publication in PLOS ONE. Congratulations! Your manuscript is now with our production department. 

Kind regards, 

on behalf of

Dr. Steve Lin 

Academic Editor

PLOS ONE